

# A longitudinal study of the diabetic skin and wound microbiome

Melissa Gardiner[1], Mauro Vicaretti[2,3], Jill Sparks[4], Sunaina Bansal[1], Stephen Bush[5], Michael Liu[1], Aaron Darling[1], Elizabeth Harry[1] and Catherine M. Burke[1]

[1] The i3 institute, University of Technology Sydney, Sydney, New South Wales, Australia
[2] Medical School, University of Sydney, Sydney, New South Wales, Australia
[3] Westmead Hospital, Western Sydney Local Health District, Sydney, New South Wales, Australia
[4] Community Nursing, Western Sydney Local Health District, Sydney, New South Wales, Australia
[5] School of Mathematical and Physical Sciences, University of Technology Sydney, Sydney, New South Wales, Australia

## ABSTRACT

**Background.** Type II diabetes is a chronic health condition which is associated with skin conditions including chronic foot ulcers and an increased incidence of skin infections. The skin microbiome is thought to play important roles in skin defence and immune functioning. Diabetes affects the skin environment, and this may perturb skin microbiome with possible implications for skin infections and wound healing. This study examines the skin and wound microbiome in type II diabetes.

**Methods.** Eight type II diabetic subjects with chronic foot ulcers were followed over a time course of 10 weeks, sampling from both foot skin (swabs) and wounds (swabs and debrided tissue) every two weeks. A control group of eight control subjects was also followed over 10 weeks, and skin swabs collected from the foot skin every two weeks. Samples were processed for DNA and subject to 16S rRNA gene PCR and sequencing of the V4 region.

**Results.** The diabetic skin microbiome was significantly less diverse than control skin. Community composition was also significantly different between diabetic and control skin, however the most abundant taxa were similar between groups, with differences driven by very low abundant members of the skin communities. Chronic wounds tended to be dominated by the most abundant skin *Staphylococcus*, while other abundant wound taxa differed by patient. No significant correlations were found between wound duration or healing status and the abundance of any particular taxa.

**Discussion.** The major difference observed in this study of the skin microbiome associated with diabetes was a significant reduction in diversity. The long-term effects of reduced diversity are not yet well understood, but are often associated with disease conditions.

Corresponding author
Catherine M. Burke,
Catherine.Burke@uts.edu.au

## INTRODUCTION

Type II diabetes is one the fastest growing chronic diseases in the world today, predicted to rise from 382 million people in 2013 to 592 million in 2035 (*Guariguata et al., 2014*). The disease is characterised by persistently elevated blood glucose levels as a result of insufficient

insulin production or insulin resistance. This leads to many serious complications affecting the heart, kidneys, eyes, blood vessels and nerves (*World Health Organisation, 2016*). The development of foot ulcers is the culmination of several of these complications, estimated to affect 15 % of diabetes sufferers (*Reiber, Boyko & Smith, 1995*). These wounds are often slow to heal, difficult to treat, and prone to infection. They have a severe impact on a patient's quality of life, and are estimated to increase the risk of lower limb amputation by 15 fold (*Australian Institute of Health and Welfare, 2008*). The cost of treating these chronic wounds is estimated at up to $13 billion annually in the US alone (*Rice et al., 2014*), and is set to rise with the increasing incidence of diabetes worldwide.

Diabetes is associated with shifts in the gut microbiota (*Karlsson et al., 2013*; *Qin et al., 2012*), and these shifts are thought to contribute to the onset of disease (*Parekh et al., 2016*; *Zhang & Zhang, 2013*). Dysbiosis of the human microbiome is increasingly recognised to play a role in many diseases, through mechanisms such as altered intestinal barrier function (*Kelly et al., 2015*), triggering or exacerbating inflammation (*Strober, 2013*) and regulation of energy metabolism (*Samuel et al., 2008*). Given the physical changes that occur in the skin as a result of diabetes, such as increased dryness and pH, and glycosylation of structural skin proteins (*Behm et al., 2012*), it is feasible that diabetes may also affect the microbiome of the skin.

As in the gut, the skin microbiome is thought to protect against infection via both competitive exclusion and direct inhibition (*Bomar et al., 2016*; *Cogen et al., 2010b*; *Iwase et al., 2010*; *Shu et al., 2013*), and have the potential to regulate skin immune function and wound healing (*Kanno et al., 2011*; *Scales & Huffnagle, 2013*). For example, the most common skin isolate, *Staphylococcus epidermidis*, has been shown to down-regulate inflammation following skin injury (*Lai et al., 2009*), and to up-regulate the production of antimicrobial peptides in the host (*Lai et al., 2010*), which work synergistically with antimicrobial peptides from *S. epidermidis* to inhibit pathogens such as *Staphyloccocus aureus* and Group A *Streptococcus* (*Cogen et al., 2010a*). Another skin commensal, *Acinetobacter lwoffii*, has been shown to protect against allergic sensitization and inflammation by promoting $T_H1$ and anti-inflammatory responses in the skin (*Fyhrquist et al., 2014*). Given the importance of the skin microbiome in preventing infection, any shifts to these communities could affect their ability to protect against infection, and may have an effect on wound healing.

The aim of this study was to determine whether there are differences in the skin microbiome between persons with diabetes and healthy controls, and whether any members of the skin microbiome in diabetes are associated with those microbes that colonise chronic wounds during wound healing. We examined a cohort of eight diabetic and eight control individuals at six time points over a 10-week period, by swabbing the skin on the soles of both feet, and collecting swabs and debrided tissue from the chronic foot ulcers of the diabetic patients. The microbial communities associated with these samples were assessed via high-throughput sequencing of the V4 region of the bacterial 16S rRNA gene.

**Table 1  Characteristics of diabetic and control cohorts.**

|  | Diabetic | Control |
|---|---|---|
| Age (years) | 68.9 ± 8.2 (58–81) | 62.8 ± 13.4 (50–81) |
| BMI | 35.4 ± 5.9 (27.2–47.1) | 28.0 ± 6.6 (20.4–37.9) |
| Males:Females | 5:3 | 2:6 |

**Notes.**

Characteristics are shown for the diabetic and control subjects in the study. Average values with standard deviations are reported, including the range in brackets.

## MATERIALS & METHODS

### Study design, ethics approval, and sample collection

Ethical approval for the study was obtained from both the University of Technology Sydney Human Research Ethics Committee (approval number 2013000170), and the Western Sydney Local Health District Human Research Ethics Committee (approval number HREC2013/9/5.3(3809) AU RED LNR/13/WMEAD/294). Diabetic individuals and control subjects provided written consent for sample collection and all subsequent analyses.

Diabetic adults ($n = 8$) (Table 1) were selected for inclusion in the study based on medical diagnosis of type II diabetes, the presence of a chronic wound on one foot (chronic wound = present for six or more weeks) and no antibiotic therapy within the previous four weeks. Three swabs were collected for each diabetic subject every two weeks for a 10 week period using sterile rayon tipped swabs (Copan) that had been pre-moistened with a sterile solution of 0.15 M NaCl and 0.1% Tween 20. Two skin swabs were collected from intact foot skin (1) adjacent to the chronic wound (skin adjacent, SA) and (2) contralateral site to the chronic wound (skin contralateral, SC). Skin swabs were collected by firmly rubbing the moistened swab over the base of the foot skin surface for a period of 30 seconds. The whole base of the foot was used to maximise the DNA yield. Skin swab samples were taken prior to any cleaning of the skin surface that routinely took place before debridement of wound tissue. Chronic wounds were cleaned by applying gauze soaked with Prontosan wound irrigation solution (B. Braun Medical, Sheffield, UK) for ten minutes prior to sharp debridement of tissue from the top of the wound (wound debridement, WD). Wound debridement samples were only taken where debridement was deemed to be necessary for the standard wound care. Wound swabs were taken after irrigation of the wound with Prontosan to remove loose tissue, using a dry swab and the Z swab method (wound swab, WS). The Z swab method was the routine method used in the clinic at the time of sampling.

Control subjects ($n = 8$) (Table 1) were recruited from Sydney, Australia. The criteria for inclusion were not to have been diagnosed as diabetic, between 50–80 years of age, and without the use of antibiotics within the previous four weeks. Skin swabs were collected from the left and right feet of control subjects as described above. Samples were taken from all participants every two weeks for a 10-week period (6 time points in total). All samples were processed for DNA on the day of collection, or stored at 4 °C until processing the next day. These storage conditions have been shown to adequately preserve the microbial profile of skin swab samples (*Lauber et al., 2010*).

## Extraction of microbial DNA from skin and wound swabs and wound debridement tissue

Genomic DNA was extracted from all skin and wound samples using the BioStic DNA extraction kit (MO BIO Laboratories, Carlsbad, CA, USA). Swab heads were cut off the plastic applicator using sterile surgical scissors into the bead beating tube from the DNA extraction kit, before addition of buffer CB1. For wound debridement tissue, the tissue was directly placed into the bead beating tube. All subsequent steps were in accordance with the manufacturer's instructions, and DNA was eluted in 50 µl of solution CB5 (10 mM Tris pH 8). The extracted DNA was quantified on a Qubit® 2.0 Fluorometer (Life Technologies, Carlsbad, CA, USA) with a Qubit® dsDNA HS Assay Kit (Life Technologies, Carlsbad, CA, USA).

## Preparation of 16S rRNA gene libraries for Illumina sequencing

A library of the V4 region of the 16S rRNA gene was prepared for Illumina sequencing from the isolated microbial DNA samples. Samples were amplified using primers based on the Caporaso et al. design (*Caporaso et al., 2012*), which were modified to include eight nt rather than 12 nt barcodes, and include a barcode on both the forward and reverse primer (V4_forward and V4_reverse; Table 2). Different barcoded primers were used for each sample. For skin samples, the V4 region was amplified from 500 pg template DNA; for wound samples template DNA started at 10 ng, but in some cases up to 50 ng was used where a PCR product was not obtained with lower amounts of template DNA. Each sample was subjected to 10 cycles of PCR with 0.5 µM each of V4_forward and V4_reverse barcoded primers in a 50 µl PCR reaction that contained $1 \times$ Taq core PCR buffer (Qiagen, Venlo, Netherlands), $1 \times$ Q solution, 250 µM dNTPs, and 1.25 U Taq DNA polymerase. Thermal cycling was carried out at 95 °C for two minutes, followed by 10 cycles of 95 °C for 15 s, 50 °C for 30 s and 72 °C for 90 s, followed by a final extension at 72 °C for five minutes. Excess primer was removed via a magnetic bead clean-up using 0.8 volume of Axygen® AxyPrep Mag beads (Corning, NY, USA) and the eluted amplicons were subjected to a further 20 cycles of PCR with 0.25 µM enrichment primers (Illumina_E_1 and Illumina_E_2; Table 2). The PCR reaction and cycling was performed as described above, except that the annealing temperature was increased to 55 °C and 20 thermal cycles were performed. Following confirmation of the PCR product on a 1% agarose gel, the amplicons were purified using Axygen® AxyPrep Mag beads (Corning, NY, USA) and quantified on a Qubit® 2.0 Fluorometer (Life Technologies, Carlsbad, CA, USA) with a Qubit® dsDNA HS Assay Kit (Life Technologies, Carlsbad, CA, USA). Equimolar (2 ng) amounts of the 16S amplicons obtained for each skin and wound sample were then pooled and the molarity of the pooled amplicons determined using a Bioanalyser High Sensitivity DNA chip (Agilent Technologies, Santa Clara, CA, USA).

## Illumina sequencing and data analysis

The PCR amplicons from 264 samples (including positive and negative controls) were sequenced over two separate runs on an Illumina Miseq using 500 cycle V2 kits. Sequences were demultiplexed using phylosift (*Darling et al., 2014*) and read pairs merged using

**Table 2  Primer sequences used in this study.**

| Primer name | Sequence 5′–3′ |
| --- | --- |
| V4_forward_1 | AATGATACGGCGACCACCGAGATCTACACAACCAGTCTATGGTAATTGTGTGCCAGCMGCCGCGGTAA |
| V4_forward_2 | AATGATACGGCGACCACCGAGATCTACACAACGCTAATATGGTAATTGTGTGCCAGCMGCCGCGGTAA |
| V4_forward_3 | AATGATACGGCGACCACCGAGATCTACACAAGACTACTATGGTAATTGTGTGCCAGCMGCCGCGGTAA |
| V4_forward_4 | AATGATACGGCGACCACCGAGATCTACACAATCGATATATGGTAATTGTGTGCCAGCMGCCGCGGTAA |
| V4_forward_5 | AATGATACGGCGACCACCGAGATCTACACACCAATTGTATGGTAATTGTGTGCCAGCMGCCGCGGTAA |
| V4_forward_6 | AATGATACGGCGACCACCGAGATCTACACACTGAAGTTATGGTAATTGTGTGCCAGCMGCCGCGGTAA |
| V4_forward_7 | AATGATACGGCGACCACCGAGATCTACACATTGCCGCTATGGTAATTGTGTGCCAGCMGCCGCGGTAA |
| V4_forward_8 | AATGATACGGCGACCACCGAGATCTACACCAACCTTATATGGTAATTGTGTGCCAGCMGCCGCGGTAA |
| V4_forward_9 | AATGATACGGCGACCACCGAGATCTACACCCTAATAATATGGTAATTGTGTGCCAGCMGCCGCGGTAA |
| V4_forward_10 | AATGATACGGCGACCACCGAGATCTACACCCTCTGATTATGGTAATTGTGTGCCAGCMGCCGCGGTAA |
| V4_forward_14 | AATGATACGGCGACCACCGAGATCTACACGAACGGAGTATGGTAATTGTGTGCCAGCMGCCGCGGTAA |
| V4_forward_16 | AATGATACGGCGACCACCGAGATCTACACGCGTTACCTATGGTAATTGTGTGCCAGCMGCCGCGGTAA |
| V4_forward_18 | AATGATACGGCGACCACCGAGATCTACACGGATGCCATATGGTAATTGTGTGCCAGCMGCCGCGGTAA |
| V4_forward_20 | AATGATACGGCGACCACCGAGATCTACACGTTGGCCGTATGGTAATTGTGTGCCAGCMGCCGCGGTAA |
| V4_forward_22 | AATGATACGGCGACCACCGAGATCTACACTGACTGCTTATGGTAATTGTGTGCCAGCMGCCGCGGTAA |
| V4_forward_24 | AATGATACGGCGACCACCGAGATCTACACTTCAGCGATATGGTAATTGTGTGCCAGCMGCCGCGGTAA |
| V4_reverse_1 | CAAGCAGAAGACGGCATACGAGATAACCAGTCAGTCAGTCAGCCGGACTACHVGGGTWTCTAAT |
| V4_reverse_7 | CAAGCAGAAGACGGCATACGAGATATTGCCGCAGTCAGTCAGCCGGACTACHVGGGTWTCTAAT |
| V4_reverse_8 | CAAGCAGAAGACGGCATACGAGATCAACCTTAAGTCAGTCAGCCGGACTACHVGGGTWTCTAAT |
| V4_reverse_9 | CAAGCAGAAGACGGCATACGAGATCCTAATAAAGTCAGTCAGCCGGACTACHVGGGTWTCTAAT |
| V4_reverse_15 | CAAGCAGAAGACGGCATACGAGATGCCTACGCAGTCAGTCAGCCGGACTACHVGGGTWTCTAAT |
| V4_reverse_16 | CAAGCAGAAGACGGCATACGAGATGCGTTACCAGTCAGTCAGCCGGACTACHVGGGTWTCTAAT |
| V4_reverse_17 | CAAGCAGAAGACGGCATACGAGATGGAGGCTGAGTCAGTCAGCCGGACTACHVGGGTWTCTAAT |
| V4_reverse_23 | CAAGCAGAAGACGGCATACGAGATTGGCGATTAGTCAGTCAGCCGGACTACHVGGGTWTCTAAT |
| V4_reverse_24 | CAAGCAGAAGACGGCATACGAGATTTCAGCGAAGTCAGTCAGCCGGACTACHVGGGTWTCTAAT |
| V4_reverse_25 | CAAGCAGAAGACGGCATACGAGATTTGGCTATAGTCAGTCAGCCGGACTACHVGGGTWTCTAAT |
| Illumina_E_1 | AATGATACGGCGACCACCGA |
| Illumina_E_2 | CAAGCAGAAGACGGCATACGA |
| V4_read_1 | TATGGTAATTGTGTGCCAGCMGCCGCGGTAA |
| V4_read_2 | AGTCAGTCAGCCGGACTACHVGGGTWTCTAAT |
| V4_index_read | ATTAGAWACCCBDGTAGTCCGGCTGACTGACT |

FLASH (*Magoc & Salzberg, 2011*). Sequences were quality filtered and processed into OTUs using USEARCH v 1.8.1 (*Edgar, 2010*) (fastq_filter command with the fastq_maxee option set to '2' to remove all sequences with two or more expected errors). Further quality filtering and operational taxonomic unit (OTU) clustering was carried out in QIIME (*Caporaso et al., 2010b*) version 1.9.0. The split_libraries.py command was used with the –l and –L options set to 240 and 260 respectively, to remove sequences shorter than 240 and longer than 260 base pairs. Sequences were clustered into OTUs at 97% similarity using the pick_open_reference_otus.py script using default settings except that singleton OTUs were removed, and the usearch61 method was used for chimera filtering.

Taxonomy was assigned to OTUs (assign_taxonomy.py) using the UCLUST method (*Edgar, 2010*) against the Greengenes (*DeSantis et al., 2006*) database pre-clustered at 97% similarity, accessed from the QIIME website (ftp://greengenes.microbio.me/greengenes_release/gg_13_5/gg_13_8_otus.tar.gz). Representative sequences from each OTU were aligned against the Greengenes alignment using Pynast (*Caporaso et al., 2010a*) (align_seqs.py), OTUs which failed alignment were filtered from the final OTU table (filter_otus_from_otu_table.py). A phylogenetic tree was built from aligned representative OTU sequences (make_phylogeny.py script) using Fasttree2 (*Price, Dehal & Arkin, 2010*), with the –r option set to midpoint for tree rooting.

Diabetic skin samples adjacent to wounds were found to be more similar to wound than contralateral skin samples (see Fig. S1), and were removed so as not to confound comparisons between diabetic and non-diabetic skin. To ensure more even sample sizes between the diabetic and non-diabetic groups, only the right foot samples were included from the non-diabetic group for all downstream analyses. Alpha diversity was calculated using Phyloseq (*McMurdie & Holmes, 2013*) for the observed number of OTUs, Chao 1 and Shannon diversity indices on data rarefied to 30,000 sequences per sample. Significance testing was carried out on alpha diversity estimates using the Wilcoxon rank sum test in R.

Initial beta-diversity analysis was carried out in QIIME on a rarefied OTU table (30K sequences per sample) using the weighted unifrac metric, and the generate_boxplots.py script used to compare unifrac distances between groups of samples. Futher beta diversity analyses, were carried out in Phyloseq, using weighted unifrac distances calculated from an OTU table with raw counts subject to variance stabilising transformation implemented in DEseq2 (*Love, Huber & Anders, 2014*) as described here (*McMurdie & Holmes, 2014*). Weighted unifrac distances matrices were also subject to principal coordinates analysis using the Phyloseq package, and significant differences in variance between groups (diabetic and control skin) were determined with PERMANOVA (adonis function) implemented in the Vegan package (*Oksanen et al., 2015*) in R, using a nested model formula (∼health/subject + subject) and the weighted unifrac distance matrix.

The Wald test for differential abundance was used as implemented in the DESeq2 package in R. Multivariate correlation analysis was carried out against OTUs and wound duration and area using Pearson scores with Bonferroni correction, and p-values were determined via bootstrapping with 100 permutations (implemented in QIIME using the observation_metatdata_correlation.py command). OTU tables were filtered to remove OTUs present in less than 10% of samples for both differential abundance and correlation tests.

A Random forest learning algorithm implemented in R (*Liaw & Wiener, 2002*) was used to determine if diabetic status could be predicted from the foot skin microbiome. Skin samples were randomly divided into two equal subsets (restricting samples from the same participant to the same subset) for training and testing of learning algorithms. The variance stabilizing transformed OTU table was filtered to include skin samples only, and to remove OTUs observed in less than 10% of samples, and used as the input matrix for the Random forest algorithm. The Random forest fitted on the training subset was created using bootstrapping of one third of the training samples with replacement. As a general

practice the rest of the samples were used as a validation set in order to decrease the risk of over-fitting associated with classification algorithms. An optimisation to minimise the out of bag error (classification error on validation data) was used to obtain the optimal number of taxonomic units accessed at each iteration of decision tree creation. Two hundred decision trees consisting of 30 OTUs evaluated at each node of the tree were created. The Random forest model was then used to predict the health status of the subjects in the test subset.

Analysis of the stability of skin microbial communities over time was carried out by comparing intrapersonal weighted unifrac distances between the diabetic and control skin samples, along with intrapersonal distances for all samples. Kruskal–Wallis tests were used to determine significant differences between groups.

Pearson's Product Moment Correlation was used to test for correlations between wound size or duration and OTU abundance in wound samples as implemented in QIIME (observation_metadata_correlation.py). $P$-values were calculated using bootstrapping with 100 permutations, and Bonferroni correction for multiple testing. Kruskal-Wallis tests for OTUs that were differentially abundant in healing vs non-healing wounds were implemented in QIIME (group_significance.py). Wounds were classified as healing or non-healing based on a reduction in wound area since the last sampling time (healing) or no change or greater wound size area since the last sampling (non-healing). OTU tables were filtered to remove OTUs present in less than 10% of samples prior to testing.

Inter-visit weighted unifrac distances were compared to the overall degree of healing (1− (final wound area/initial wound area)) using the lm function of the stats package in R.

Quality filtered sequence data has been deposited in the European Nucleotide Archive under study accession number PRJEB17696. A script containing the code used to process the data in R is provided as supplementary data, along with all the necessary input files, including OTU table and phylogenetic tree.

## RESULTS

### Cohort characteristics

The diabetic cohort ($n = 8$) consisted of five males and three females, with an average age of 68.9 ± 8.2 (range 58–81), average BMI of 35.4 ± 5.9 (range 27.2–47.1), and all had at least one foot ulcer which had been present for a average time of 9.1 ± 8.4 months (range 1.5–24 months). All wounds were neuropathic, with the exception of Patient 6 where the wound was ischemic. Two of the eight wounds healed during the course of sampling. Wounds were dressed with either Allevyn foam (Smith and Nephew) to promote moist wound healing, Zetuvit dressing (Hartmann) to remove excess wound exudate, Inadine antimicrobial dressing (10% povidone-iodine) (Johnson and Johnson), or Acticoat flex (antimicrobial silver coated) (Smith and Nephew), as deemed appropriate by the treating podiatrist or wound care nurse. All wounds were located on the plantar aspect of the foot. Details of the specific location of each wound, along with size and treatment over time and are provided in Table S3.

The control cohort ($n = 8$) consisted of 2 males and 6 females, with an average age of $62.8 \pm 13.4$ (range 50–81), average BMI of $28.0 \pm 6.6$ (range 20.4–37.9), and did not have wounds present on the feet.

## Sample processing, 16S PCR and sequencing

A total of 242 samples were collected from the diabetic and control cohorts, including 170 skin swabs (85 diabetic and 85 control), 40 wound swabs and 32 wound debridement samples. Full details for samples collected at each time point for each participant can be found in Table S1 (diabetic participants) and Table S2 (control participants).

DNA yields obtained from diabetic skin swabs varied from 0.51 to 600 ng, with a median of 8.5 ng. Three skin samples did not yield enough DNA to be measured by the Qubit assay, however 16S rRNA gene PCR products were still obtained. Control skin sample DNA yields ranged from 0.5 to 41.7 ng (median 5.55), with 20 samples falling below the detection limit of the Qubit assay (<5 pg/µl). Of these 20 samples, PCR products were obtained for all but 3. DNA yields from wound swab samples ranged from 15 ng to 5.6 µg (median 760 ng) and for wound debridement samples ranged from 170 ng to 5.8 µg (median 1.2 µg). One wound swab sample did not yield enough DNA to be detected. Negative control swabs ($n = 4$) did not yield enough DNA to be detected, and also did not yield detectable PCR products.

PCR products from the V4 region of the 16S rRNA gene were obtained for 257 of the 273 samples collected. Repeated attempts were made with increased amounts of template for those samples that did not initially yield a PCR product, however no PCR product was obtained (detailed in Tables S1 and S2). Amplicons from the remaining 257 samples were pooled and paired-end sequenced over two separate MiSeq runs with V2-500 cycle kits. Sample from four diabetic and four control subjects were sequenced in each run (Table S4). A median coverage of 73,599 sequences per sample was obtained (minimum 1,683, maximum 297,817). Negative controls (two blank swab and 2 no DNA PCR controls) had between 1,508 and 27,840 sequences assigned. The final sequencing coverage obtained for each sample can be found in Table S4 . Because negative control samples contained taxa that are similar to those found on skin (e.g., *Staphylococcus, Corynebacterium* and *Acinetobacter*) specific taxa were not removed from the data, rather samples with less than 30,000 sequences ($n = 5$) were removed from the analysis, based on the highest level of sequencing reads obtained from negative controls. A PERMANOVA test was run on a weighted unfrac distance matrix generated from variance stabilising transformed counts to assess the amount of variance attributable to the two different sequencing runs, (run + subject). Sequencing run was a significant factor accounting for 3.0% of the variance ($p < 0.001$), while inter-individual differences accounted for 34.5% ($p < 0.001$).

## The microbiome of diabetic skin is less diverse than control skin

Diversity in all three groups was significantly different for observed richness, Chao1 and Shannon diversity indices (likelihood ratio test, $p < 0.01$). Diabetic skin was significantly less diverse than control skin for richness and Chao1 indices (Wilcoxon rank sum test, $p < 0.01$) (Fig. 1). Control skin had a median of 998.5 observed OTUs, compared to 435 for diabetic skin. Wounds were also significantly less diverse than diabetic skin with a median of 145 observed OTUs.

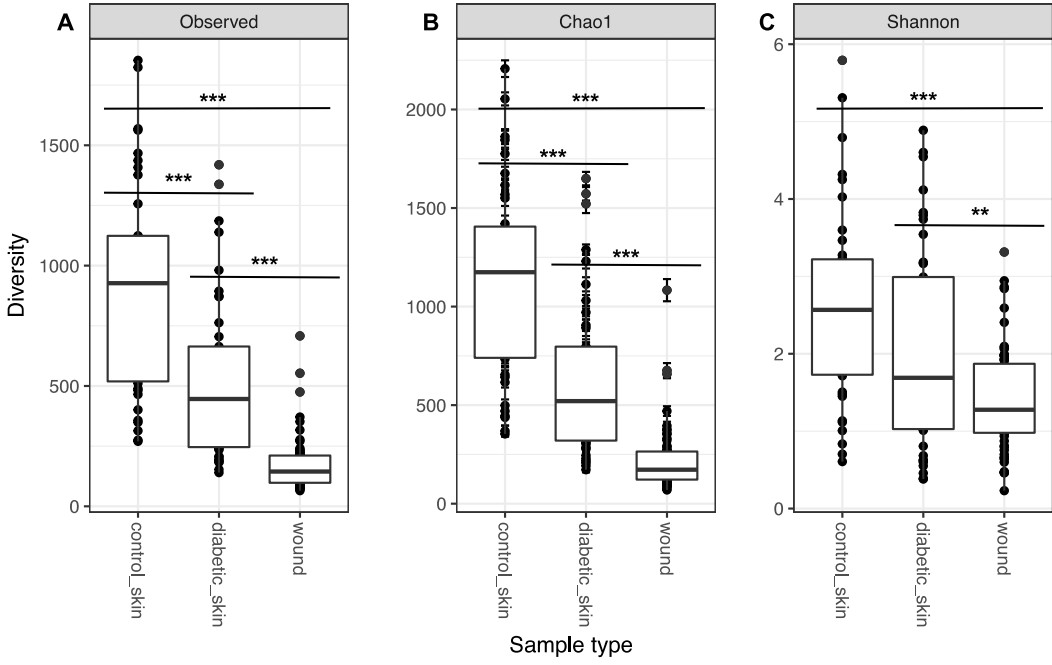

**Figure 1  Alpha diversity of skin and wounds.** Box plots of 3 different alpha diversity measures, (A) observed number of OTUs or richness, (B) the Chao I estimator, and (C) the Shannon index, based on OTUs clustered at 97% similarity for control skin, diabetic skin and diabetic wounds. Significant differences are indicated by asterix * = $p < 0.05$, ** = $p < 0.01$ *** = $p < 0.001$.

## The skin microbiome is significantly different between diabetic and control subjects

Skin microbial communities overall were significantly different between diabetic and control skin (Fig. 2). A clear distinction can be observed between the sample types, and this was confirmed by a PERMANOVA test (~health/subject + health), where health (diabetes vs control) was a significant factor accounting for 11.7% of the variance ($R^2 = 0.117$, $p = 0.001$). Subject (inter-individual differences) was the most significant factor accounting for 34.6% of the observed variance ($R^2 = 0.346$, $p = 0.001$).

## Abundant taxa from skin are similar between persons with diabetes and healthy controls

Despite the clear distinction between diabetic and control skin in the PCoA plot above, the most abundant taxa from both groups were similar. Foot skin communities from diabetic skin were dominated by the genera *Staphylococcus*, followed by *Acinetobacter and Corynebacterium*, then unclassified Enterobacteriacea. Control skin was dominated by the genera *Staphylococcus*, followed by *Acinetobacter, Kocuria, Corynebacterium* and *Micrococcus*, (Fig. 3).

To determine which OTUs were contributing to the significant difference detected in the PERMANOVA analysis, the Wald test as implemented in the DESeq2 package (*Love, Huber & Anders, 2014*), was carried out. Sixty-nine OTUs were identified as significantly different in abundance (adjusted $p < 0.05$), all with an average abundance of less than 1%. A full list

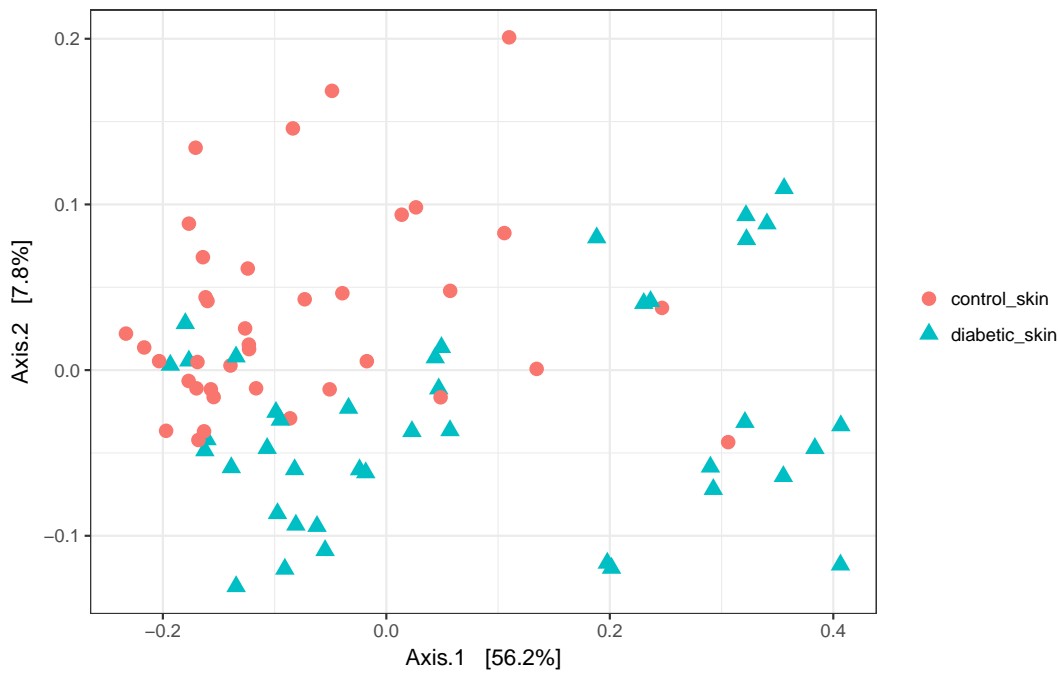

**Figure 2** **Principal coordinates analysis of diabetic and control skin samples.** Distances are based on the weighted unifrac metric, calculated using raw counts subjected to a variance stabilising transformation.

of the results can be found in Table S5. Similar results were found when re-running the analysis at the Genera level, with 24 genera identified as significantly different, but all at an average relative abundance of less then 1% (Table S6).

### The foot skin microbiome may predict diabetic status

Despite only low abundance OTUs showing significant differences between diabetic and non-diabetic skin, a Random Forrest classifier was able to predict diabetic status from the foot skin microbiome. The model achieved an overall accuracy of 85.0%, with a sensitivity of 79.2%, and specificity of 93.8%. The negative predictive value (75.0%) was lower than the positive predictive value (95.0%). The classifier's Gini index provided a list of 106 OTUs that were important in the classification task (Table S7); the majority were low abundance OTUs (103 OTUs < 1% average relative abundance), and the majority of these were more abundant in control than diabetic skin (75 OTUs).

### Stability of the diabetic skin microbiome over time

Longitudinal analysis of the skin microbiome over time showed a trend of lower stability for diabetic skin than non-diabetic skin (Fig. 4), however this difference did not reach significance ($p = 0.09$), while both control and diabetic skin intrapersonal differences over time were significantly smaller (i.e., more stable) than inter-individual differences ($p < 0.05$).

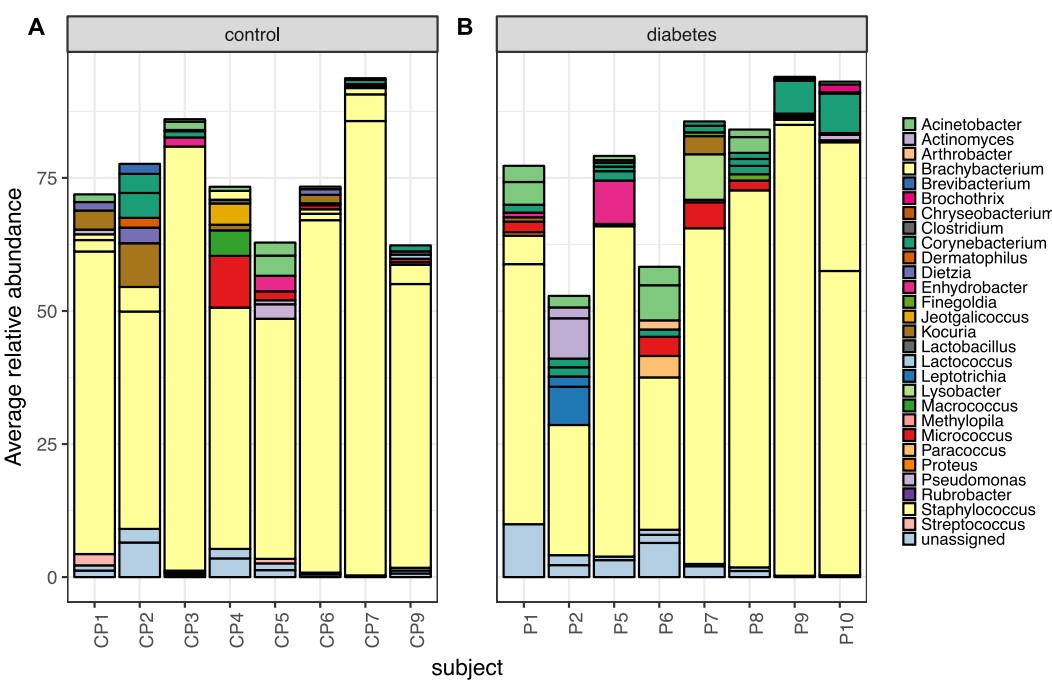

**Figure 3** **The top 10 most abundant OTUs in diabetic and control skin per subject.** The top 10 most abundant OTUs in (A) control and (B) diabetic skin per subject. Average abundances per person were calculated from data rarefied to 30,000 sequences per sample. Genus assigned taxonomy is indicated in the legend, individual OTUs of the same genera are indicated with black lines.

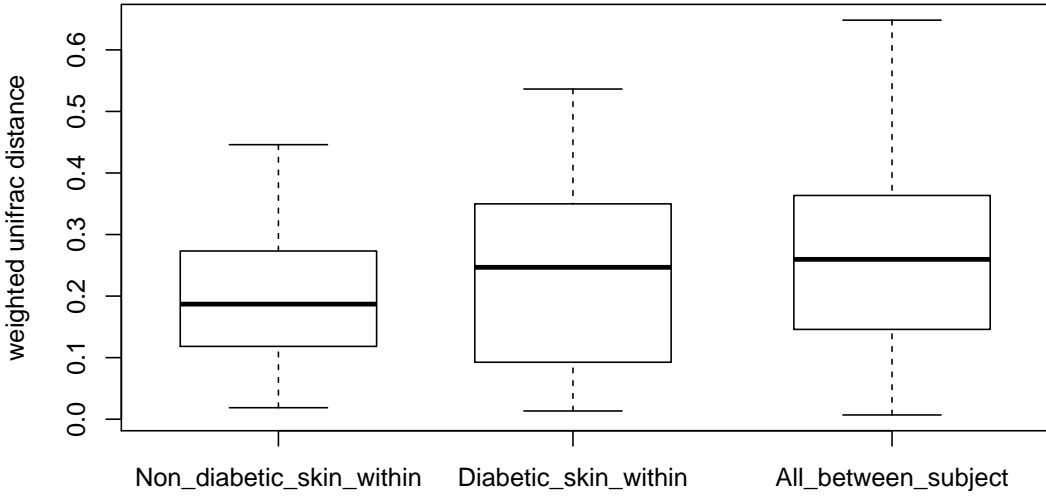

**Figure 4** **Boxplots of intra-individual differences over time in diabetic and non-diabetic skin microbial communities.** Inter-individual distances are also shown for comparison. The stability of non-diabetic skin was higher (i.e., lower distances over time) than for diabetic skin, however this difference did not reach significance. (Kolmogorov–Smirnov test, $p = 0.09$).

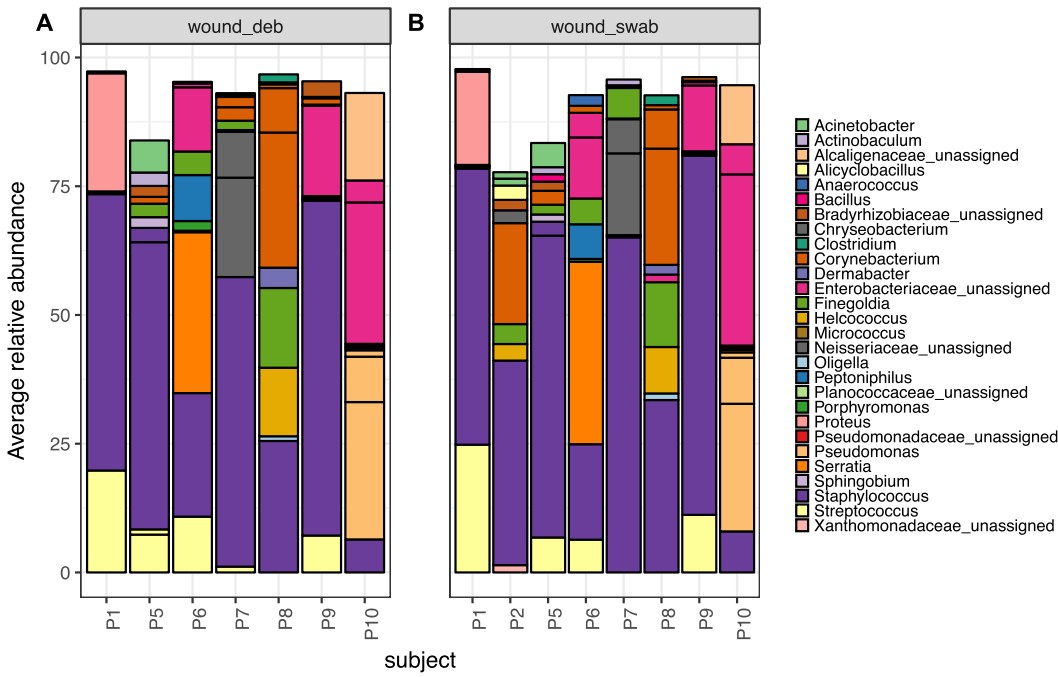

**Figure 5** **The top 10 abundant OTUs in wounds per subject.** The top 10 abundant OTUs per subject in diabetic (A) wound debridement and (B) wound swab samples. Average abundances per group were calculated from data rarefied to 30,000 sequences per sample. Genus assigned taxonomy is indicated in the legend, or family level where genus was unassigned. Individual OTUs of the same genera are indicated with black lines.

## Microbiota of chronic diabetic wounds overlap with skin and differ between patient

Wound swab and debridement samples were similar in taxonomic composition, and the top ten OTUs from all wounds per patient are shown in Fig. 5. The most abundant OTU detected in wounds was also the most abundant OTU found on skin, *Staphylococcus sp.* (OTU 1084865), and was present in the wounds of all eight patients. Other skin associated OTUs found in wounds included *Corynebacterium* (OTU 1011712), which was in the top 10 OTUs in six out of eight patient's wounds.

The Wald test for differential abundance between diabetic skin and wounds identified four OTUs that were significantly more abundant across all wounds (two classified as Enterobactericaeae, one as *Serratia* and one as *Finegoldia*). The complete list of results can be found in Table S8.

The top 10 OTUs in wounds per patient over time are shown in Fig. 6. Of the eight wounds, six are dominated by the most abundant skin OTU, at the majority of time points measured (Staphylococcus OTU 1084865). Only Patients 6 and 10 showed wound profiles dominated by non-skin associated taxa across the time period surveyed. No significant correlations were found between any abundant OTUs (average abundance > 1%) and wound duration or healing status. No significant correlation was found between the overall degree of wound healing, and inter-visit weighted unifrac distances in individual wounds (Fig. S2, $p = 0.29$). However, some interesting observations were made that correlated to

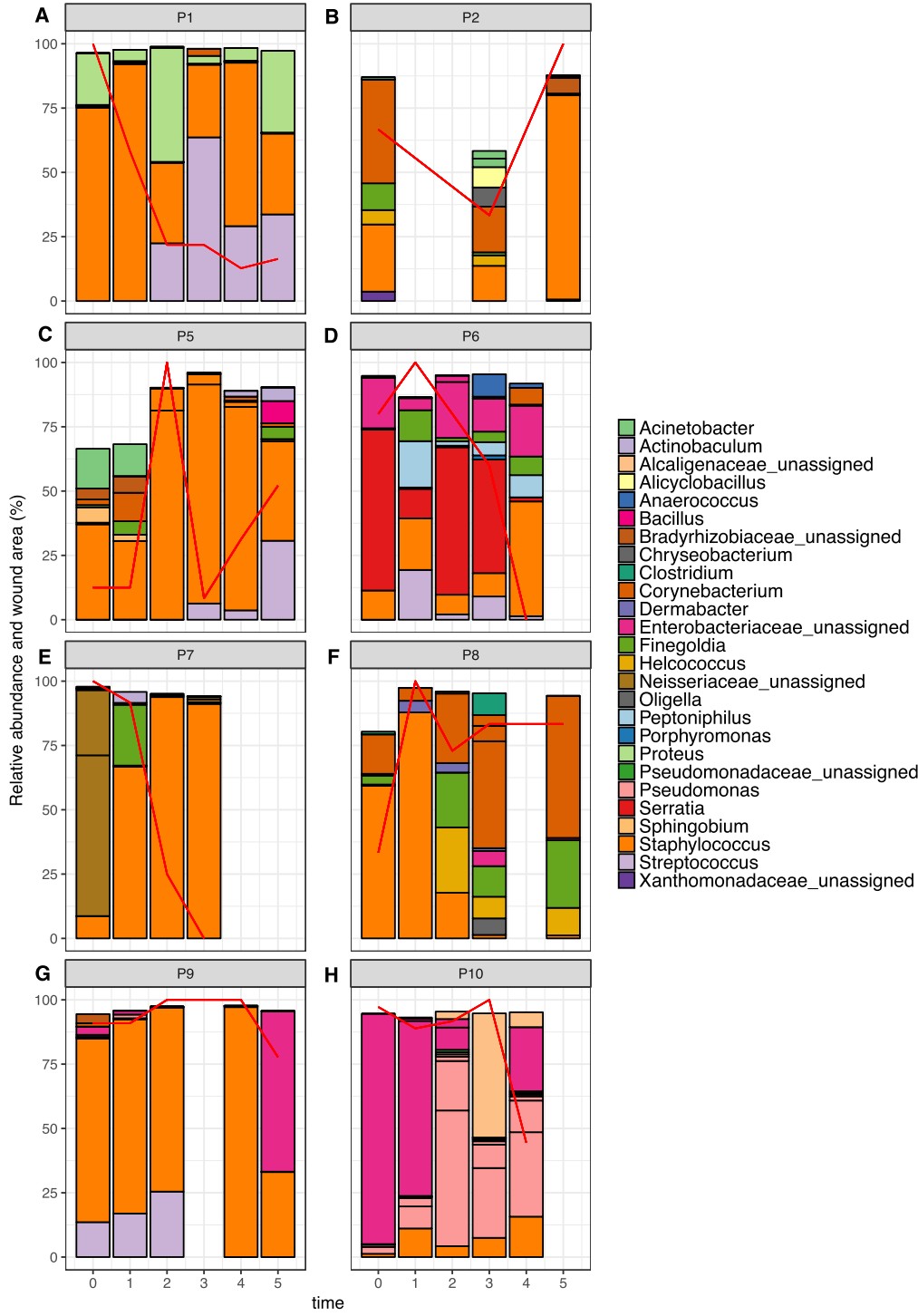

**Figure 6 Relative abundance of the top 10 OTUs per patient over time.** Patients 1–10 are represented individually in (A–H). Wound area is overlaid as a red line and is represented as a percentage of the largest wound area measured over time. Relative abundances were calculated from data rarefied to 30,000 sequences per sample. Genus assigned taxonomy is indicated in the legend, or family level where genus was unassigned. Individual OTUs of the same genera are indicated with black lines.

clinical events. For example, the wound of Patient 6 had been present for 24 months at the start of the study. It was dominated by Enterobacteriacaea and showed little healing until time point 3, which coincided with an angioplasty procedure to improve blood flow to the foot. This was followed by resolution of the wound within two weeks. When Patient 7 presented to the clinic, the wound had been present for 12 months, and was dominated by an OTU from the Neisseriaceae family. Following the standard treatment of debridement and wound dressing, rapid healing was observed, as well as a shift to a community dominated by the most abundant skin OTU.

## DISCUSSION

This study aimed to compare the skin microbiome between persons with diabetes and healthy control individuals over time. We additionally sought to characterise the wound microbiota in diabetic foot ulcers over time and determine if any members of the skin microbiome were correlated to the wound microbiome or wound healing.

The microbiome from diabetic skin was significantly different to that of control skin, however this difference was not driven by the most abundant members of the skin community. The top 10 most abundant OTUs per person were similar in abundance and not significantly different between groups. Many low abundance OTUs were identified as significantly different, with the vast majority of these being more abundant in control skin. One limitation of this study is that, although commonly used in microbiome studies (*Cope et al., 2017*; *David et al., 2014*; *Halfvarson et al., 2017*; *Smith et al., 2016*), the V4 region of the 16S rRNA gene does not allow differentiation between *Staphylococcus aureus* and other *Staphylococcus* species found on skin, such as *Staphylococcus epidermidis* (*Conlan, Kong & Segre, 2012*). Additionally, the V4 primers have mismatches that prevent detection of *Propionibacterium*, an important genera in the skin microbiome (*Kuczynski et al., 2011*). The clinical consequences of these organisms may be important, and this should be taken into consideration for the experimental design of future studies (*Gohl et al., 2016*; *Meisel et al., 2016*).

We observed a significant reduction in alpha diversity and a trend of decreased stability (non-significant) of diabetic skin microbiomes compared to non-diabetic skin. This is in contrast to a previous study of diabetic skin (*Redel et al., 2013*) where the opposite result was observed. It is possible that changes to the skin environment associated with diabetes, such as increased pH (*Yosipovitch et al., 1993*) advanced glycation end products in the skin matrix (*Gkogkolou & Bohm, 2012*), or increased levels of skin inflammation (*Tellechea et al., 2013*) could drive a decrease in diversity. It is also possible that activities associated with diabetes, such as increased exposure to antibiotics (*Mor et al., 2016*), contribute to the observed effect despite our attempts to control for recent antibiotic exposure as a confounding variable. Another limitation of the current study is the small sample size, and as such this result should be confirmed on a larger cohort.

If skin microbiome diversity is depleted in people with diabetes, what are the implications for the health of diabetic skin? While in some body sites an increase is microbial diversity is associated with disease states, particularly the vagina (*Van de Wijgert et al., 2014*), decreased

diversity of the microbiome has frequently been correlated with disease and inflammation in the skin (*Alekseyenko et al., 2013*; *Ellebrecht et al., 2016*; *Seite et al., 2014*; *Williams & Gallo, 2015*), gut (*Giloteaux et al., 2016*; *Sze & Schloss, 2016*) and airways (*Yu et al., 2015*). However it is not known whether decreased diversity in these sites is a cause or merely an indicator of inflammation. Diversity is commonly used as an indicator of ecosystem health, with decreased diversity typically signalling a disturbed and less resilient state (*Oliver et al., 2015*). In the context of the human skin microbiome, decreased diversity could allow potential pathogens to overgrow, and these may be capable of triggering inflammation and triggering or exacerbating a disease state. Alternatively, inflammation could be triggered by genetic and environmental factors, and the inflammation itself could drive down bacterial diversity by creating an inhospitable growth environment.

Patients with diabetes enrolled in this study had no exposure to antibiotics within the previous four weeks, so as not to confound the comparison between diabetic and control skin. This meant that the foot ulcers analysed in this study were considered to be clinically non-infected wounds. No significant correlations were found between any OTU in diabetic skin or wounds with wound size, duration or healing status. This is possibly due to the small sample size, as a previous study found correlations between the relative abundance of specific bacterial taxa and ulcer duration and depth (e.g., *Staphylococcus* was negatively correlated with wound duration) (*Gardner et al., 2013*). Another possible limitation of this study is the use of the z-swab method which samples across the entire wound base regardless of size, as this will possibly increase heterogeneity with increasing wound size.

A recent longitudinal study of wounds found a negative correlation between wound microbiota stability and time to heal (*Loesche et al., 2017*). We did not find any such correlation here when comparing degree of healing to between visit weighted unifrac distances (Fig. S2), although again our sample size was smaller, as was the length of time patients were followed.

The overall composition of the diabetic wound microbiota described here is in agreement with a survey of 910 chronic diabetic foot ulcers, where a dominance of *Staphylococcus*, as well as *Pseudomonas*, *Corynebacterium*, *Streptococcus* and *Finegoldia* (among others) was found (*Wolcott et al., 2016*). *Gardner et al. (2013)* found that diabetic ulcers clustered into three types, depending on the dominant taxa in the wounds, which were *Staphylococcus*, *Streptococcus*, or a mixture of anaerobic bacteria or Proteobacteria. Similar results were found in a later study where two wound clusters were dominated by either *Staphylococcus* or *Streptococcus*, and genera such as *Corynebacterium* and *Finegoldia* were frequently observed (*Loesche et al., 2017*). These same genera were observed in most wounds here, while other genera such as *Serratia* and *Proteus* were specific to individuals.

Other studies of diabetic foot ulcers have reported contrasting results, such as a dominance of *Corynebacterium* (*Dowd et al., 2008*), while a recent study found that *Staphylococcus* were common in new ulcers, but not in recurring ulcers (*Smith et al., 2016*). One trend that was consistent across several studies was that the microbial profile from diabetic ulcers was variable, with no one typical diabetic ulcer microbiota apparent.

## CONCLUSIONS

The major effect associated with diabetes observed here was a significant reduction in the diversity of the skin microbiome. The cohort of this study was small, and these observations should be verified in a larger study. The long-term effects of reduced diversity are not yet well understood, but low diversity continues to be linked to disease and poor health outcomes (*Hua et al., 2016*; *Miller et al., 2016*; *Rook, 2013*). One possible effect is increased infection susceptibility (*Seto et al., 2014*), and it is intriguing to consider whether decreased skin microbiome diversity could be contributing to the high incidence of skin and wound infections associated with this disease (*Peleg et al., 2007*). There are, of course, many other well-documented factors such as immune dysfunction that can contribute to an increased rate of infections (*Geerlings & Hoepelman, 1999*); however, the skin microbiome may be an as yet unconsidered contributor to this phenomenon.

## ACKNOWLEDGEMENTS

We would like to acknowledge Westmead Hospital and the Podiatry Clinic Staff who kindly assisted in the collection of samples. We also acknowledge Professor Mark Morrison and Dr Paraic O. Cuiv for helpful discussions about the study design.

### Funding

Funding for this study was provided by internal grants from the University of Technology Sydney. The funders had no role in study design, data collection and analysis, decision to publish, or preparation of the manuscript.

### Grant Disclosures

The following grant information was disclosed by the authors:
University of Technology Sydney.

### Competing Interests

The authors declare there are no competing interests.

### Author Contributions

- Melissa Gardiner performed the experiments, analyzed the data, wrote the paper, prepared figures and/or tables, reviewed drafts of the paper.
- Mauro Vicaretti, Jill Sparks and Michael Liu performed the experiments, reviewed drafts of the paper.
- Sunaina Bansal and Stephen Bush analyzed the data, reviewed drafts of the paper.
- Aaron Darling contributed reagents/materials/analysis tools, reviewed drafts of the paper.
- Elizabeth Harry conceived and designed the experiments, contributed reagents/materials/analysis tools, reviewed drafts of the paper.

- Catherine M. Burke conceived and designed the experiments, performed the experiments, analyzed the data, contributed reagents/materials/analysis tools, wrote the paper, prepared figures and/or tables, reviewed drafts of the paper.

## Human Ethics

The following information was supplied relating to ethical approvals (i.e., approving body and any reference numbers):

Ethical approval for the study was obtained from both the University of Technology Sydney Human Research Ethics Committee (approval number 2013000170), and the Western Sydney Local Health District Human Research Ethics Committee (approval number HREC2013/9/5.3(3809) AU RED LNR/13/WMEAD/294). Diabetic individuals and control subjects provided written consent for sample collection and all subsequent analyses.

## Data Availability

Quality filtered sequence data has been deposited in the European Nucleotide Archive under study accession number PRJEB17696.

## Supplemental Information

Supplemental information for this article can be found online at http://dx.doi.org/10.7717/peerj.3543#supplemental-information.

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
