# Peer review of "A longitudinal study of the diabetic skin and wound microbiome"

_PeerJ, doi:10.7717/peerj.3543_

## Round 0.1 · original submission · Major Revisions

· Academic Editor

Major Revisions

Dear Catherine,

Thank you for submitting your paper with PeerJ. We received the comments from the referees and these follow on this mail. I am glad to tell you that both the referees support the publication of the paper although there are some comments that require your attention. I would ask you to revise your manuscript accordingly: I expect a suitable revised manuscript to be accepted for publications. Please be sure to address all the points in full. I am looking forward to receiving your amended version, my best regards

Reviewer 1 ·

Basic reporting

1. Some relevant literature is either not cited or mis-interpretted. In the discussion, the Wolcott 2016 WRR paper is referred to as not detecting correlations between specific taxa and wound healing. Wolcott et al did not collect outcome data in this study. The authors do not cite a recently published paper that did find relationships between the wound microbiome and outcome (Loesche et al 2017 Journal of Investigative Dermatology). This was also a longitudinal study of DFU microbiome and seems to be the most relevant for the authors to compare their findings.
2. Another misinterpretation/overgeneralization is in the discussion when considering diversity and “health”. It is definitely not an absolute that diversity = health state. Diverse vaginal microbial communities are pretty much synonymous with disease state. Similarly, the extensive skin microbiome literature shows that sebaceous skin sites lack diversity—that does not mean that they are unhealthy but rather are selected for in the microenvironment.
3. Since the cohort is so small, the authors should show data for all subjects in Figures 3 and 5. This would increase the informativeness of the figures. Stacked bar charts are a poor way to represent this data since variability is not captured.
4. The authors should avoid the use of the label “diabetics”. “Persons with diabetes” is a more appropriate practice.

Experimental design

1. Essential control data are missing. The sequencing data on the negative control swabs are not shown/reported. What taxa are present in those swabs? There should also be a control where the contamination in the Prontosan wound cleansing solution is reported. Another essential but missing control is a positive control (for example a mock community) to assess variability between sequencing runs.
2. All analysis shown is at the OTU level. It would be worthwhile to analyze data at the phylotype/taxonomy level. Sequencing error, especially in short reads like V4, can inflate OTU counts and diversity.
3. Were singleton OTUs removed?
4. The authors state themselves that the V4 region is poor at speciating Staph. Why do they include species-level data then when reporting relative abundances? This is misleading if the classifications cannot be confidently assigned.
5. The authors should report what type of DFU were in the cohort: Neuropathic? Ischemic? Characteristics such as wound location, wound size, wound depth should also be reported and controlled for in the analysis where appropriate since these variables have previously been shown to associate with wound microbiota.
6. How were the DFUs treated other than debridement and washing? Offloading? Topicals? Dressings? Did they all receive the same treatment?

Validity of the findings

The validity of the findings as they stand is questionable. To increase the validity, the authors should:
1. Report control data (see Experimental Design section, point 1)
2. Control for wound variables (see Experimental Design section, point 4)
3. Line 279-281: This statement is seemingly false just looking at Figure 3. Clearly Moraxellaceae, Micrococcaceae, and Enterobacteraceae are more abundant than Corynebacterium. Actinobacteria is a phylum and as such is not even represented in Figure 3 (though Corynebacterium is an Actinobacteria).

Reviewer 2 ·

Basic reporting

The language used throughout is professional and without major errors. However, the references to primary literature are lacking. While the authors make reference to studies on diabetic versus healthy skin they have only selectively chosen diabetic foot ulcers to reference. There have been numerous studies published that are relevant to this paper and should be sufficiently discussed, including Loesche et al. 2016. JID, Gardner et al. 2013. Diabetes, Smith et al. 2016. BMC Microbiology; Price et al. 2009. PloS One etc.).

Experimental design

-L254 you state that the negative control swabs did not yield PCR products but on L266 you say that subsampling at 30000 was chosen because of the number of reads obtained from negative controls. Please clarify.
-Could you please clarify how the percent variance for diabetes vs. control and inter-individual differences was calculated in the methods? This is interesting but not is what is presented in Figure 2, so please just clarify how it was calculated.
-Could you also clarify in the methods how the longitudinal stability metric was calculated (I am assuming it is the unifrac distance calculated between samples from a single patient at different timepoints but this is not defined in the methods).

Validity of the findings

-The findings would be better presented discussed in the context of the literature on the microbiome of diabetic foot ulcers. The one paper cited (Wolcott et al. 2016) was mixed etiology.
-The authors cite Meisel et al. 2016 a limitation to speciating Staphlyococcal spp. however that paper also says that the 16S V4 region misses Propionibacterium spp. The authors should also discuss this and the potential implications of missing Propionibacterium from their analysis as it is a normal part of the healthy skin microbiome and can also be found in diabetic foot ulcers.
-The authors should discuss potential confounding factors from sample size. The z-swabbing technique swabs across the entire wound bed (from wounds of different sizes, even in a single wound over time) and this is also a very different area from the entire base of the foot. What are the implications of this in terms of total diversity detected etc.?

Additional comments

No other comments.

---

## Round 0.2 · accepted · Accept

· Academic Editor

Accept

Thank you for submitting an amended version of your paper. I have checked the revision against the review comments and I am pleased to inform you that your work is accepted for publication within PeerJ.